# The Effects of Active Self-Correction on Postural Control in Girls with Adolescent Idiopathic Scoliosis: The Role of an Additional Mental Task

**DOI:** 10.3390/ijerph17051640

**Published:** 2020-03-03

**Authors:** Elżbieta Piątek, Michał Kuczyński, Bożena Ostrowska

**Affiliations:** 1Faculty of Physiotherapy, University School of Physical Education in Wroclaw, Poland, Ul. Paderewskiego 35, 51-612 Wrocław, Poland; bozena.ostrowska@awf.wroc.pl; 2Faculty of Physical Education and Physiotherapy, Opole University of Technology, ul. Prószkowska 76 b, 45-758 Opole, Poland; michal.kuczynski@awf.wroc.pl

**Keywords:** adolescent idiopathic scoliosis, active self-correction, postural control

## Abstract

Due to balance deficits that accompany adolescent idiopathic scoliosis (AIS), the potential interaction between activities of daily living and active self-correction movements (ASC) on postural control deserves particular attention. Our purpose was to assess the effects of ASC movements with or without a secondary mental task on postural control in twenty-five girls with AIS. It is a quasi-experimental within-subject design with repeated measures ANOVA. They were measured in four 20-s quiet standing trials on a force plate: no task, ASC, Stroop test, and both. Based on the center-of-pressure (COP) recordings, the COP parameters were computed. The ASC alone had no effect on any of the postural sway measures. Stroop test alone decreased COP speed and increased COP entropy. Performing the ASC movements and Stroop test together increased the COP speed and decreased COP entropy as compared to the baseline data. In conclusion, our results indicate that AIS did not interfere with postural control. The effects of the Stroop test accounted for good capacity of subjects with AIS to take advantage of distracting attentional resources from the posture. However, performing both tasks together exhibited some deficits in postural control, which may suggest the need for therapeutic consultation while engaging in more demanding activities.

## 1. Introduction

Adolescent idiopathic scoliosis (AIS) has been defined as a three-dimensional deformity of the spine and trunk occurring in healthy pubertal children [1]. The prevalence of AIS with a Cobb angle of >10° is approximately 2.5% in the general population, so it is the most common deformity of the spine in the maturing population [2].

From our own observations as physiotherapists, we consider that teaching active self-correction (ASC) is one of the most important tasks in the rehabilitation program. This is with the agreement of the International Scientific Society on Scoliosis Orthopaedic and Rehabilitation Treatment (SOSORT) recommendations [3]. Patients with AIS should perform ASC during their daily activities [3,4,5]. This assignment is very problematic for them, and we can see that during concurrent activities of daily living, performing ASC may interfere with posture control. According to Weiss et al. [5], the ability to adapt and maintain the properly corrected body posture whilst completing activities of daily living is crucial. It is one of the factors determining the effectiveness of corrective programs concerning the improvement of body posture.

Recently, Piątek et al. [6] reported that performing ASC resulted in a significant backward excursion of the center of pressure (COP) mean position with the concomitant increase in the COP fractality and frequency in the mediolateral (ML) plane. These results seem to account for desirable changes in postural control that were generated by learned ASC movements. In particular, they reflect the tendency of individuals with AIS to optimize their gravity line alignment and their adequate resources of available postural strategies, which are necessary to cope with novel postural challenges. However, this apparent improvement was achieved at the cost of lower automaticity, i.e., higher attentional involvement in postural control in the anteroposterior (AP) plane.

Dual-tasking is common in daily life. One normally needs to maintain postural control while performing one or more other concurrent tasks such as walking while talking [7] or keeping a stable standing position maintained during solving mental tasks. An individual’s attention resources and information processing capacity are presumably limited and must be shared among all the tasks being concurrently performed [8]. The everyday routine of AIS patients often requires the simultaneous performance of ASC with other activities of daily living; therefore, a possibility of conflict arises that may adversely affect posture control, as indicated in the recent study by Piątek et al. [6]. Accordingly, when two tasks are being performed at the same time, the performance of one or both can be impaired if together, they require attention that exceeds an individual’s capacity. Chang et al. [9] confirm this in their studies of adolescents with idiopathic scoliosis.

The aim of this study was to investigate the effects of three additional tasks on postural control in patients with AIS: (1) performing ASC; (2) an additional mental task; and (3) performing both these tasks simultaneously. Owing to the lack of reports on the relationship between AIS and the effect of attentional resource allocation, and the relatively high physical activity of the AIS subjects, we hypothesized an advantageous effect of the mental task on postural control. Similarly, the relatively well-learned ASC movements should not interfere with postural control. However, combining these tasks together may adversely affect postural performance or strategies. In patients with AIS, it was expected to gain new insights into the ASC performance during activities of daily living. That might shed new light on the modalities responsible for the emergence and sustainability of automaticity in postural control.

## 2. Materials and Methods

### 2.1. Participants

Twenty-four patients with AIS participated in this study. The subjects were 100% post-menarche females (aged 11–15.5 years) with a diagnosis of AIS from a local therapeutic rehabilitation center. All the girls had normal vision. The inclusion criteria for the participants were diagnosis of AIS by an independent physician and having received conservative treatment in the form of physiotherapeutic scoliosis-specific exercise (PSSE) for at least three to a maximum of five months. Patients were excluded for a history of spine surgery, musculoskeletal or neurological disease, back pain, or any spinal pathology not comorbid with AIS.

All patients knew their own ASC movements, which reduced scoliotic curves. These included different types of movements: (1) controlled self-elongation, having regard to the sagittal plane; (2) correction of the primary curve in the frontal plane; (3) correction of contiguous curves in the frontal plane; (4) correction of the primary curve in the horizontal plane. The biometric characteristics, including scoliotic curvature details, are presented in Table 1.

Written informed consent was obtained from all participants and their parent(s) or legal guardian(s) prior to their participation. The study goals, procedures, and methods were explained in full, and the subjects were informed that they could withdraw at any time. The study was approved by the Senate Research Ethics Committee at the University School of Physical Education in Wroclaw, Poland (approval number: 35/2016).

### 2.2. Methods

Postural control was assessed with the eyes open on a Kistler force platform (Kistler 9281CA, Winterthur, Switzerland). Two-dimensional horizontal coordinates of the COP data were recorded for 20 s at a sampling frequency of 100 Hz.

Each participant performed four quiet standing trials on a foam pad placed on the platform (5 cm thick foam). These included: (1) QST: standing upright with a neutral and comfortable stance with the arms relaxed at the sides; (2) ASC: standing upright with autocorrection, where on the “correction” command, the participant performed ASC; (3) QST + MT: QST with a modified color-word Stroop test as an additional mental task; and (4) ASC + MT: dual-task: performing ASC with a modified color-word Stroop test as an additional mental task. The order of the four tasks was randomized. The feet position (5 cm apart) was marked on the surface to ensure repeatability across trials and participants. The participants were to focus their gaze on a computer screen at the eye level at a distance of 1.5 m and stand as motionless as possible. The Stroop test was projected onto the screen in the form of colored words whose color was different from what they read (for instance, the word “red” was projected in green ink). During this task, the participants were required to name the color of the ink instead of reading the word [10]. The sole purpose of the Stroop test was to distract their attention from the postural task. Data acquisition began when the subject signaled that they were ready.

Relevant COP outcome measures were determined separately for the ML and the AP direction. On the basis of the COP recordings, spatial and temporal COP parameters were computed. These were:COP variability [mm] – standard deviation of COP displacement from mean COP;COP mean speed [mm/s] – COP excursion divided by trial time;COP fractal dimension – a non-linear dynamic parameter of COP where the greater the fractal dimension, the better the postural system adapts to changes;COP sample entropy – a non-linear dynamic parameter of COP where greater entropy (higher COP irregularity) suggests less attentional resources devoted to balancing maintenance (greater automaticity) [11,12];incremented COP sample entropy – computed for the velocity of the COP displacements [13].

### 2.3. Statistical Analysis

The Statistica 12.0 software package (StatSoft, Tulsa, OK, USA) was used to carry out all statistical analyses. The data met the criteria of the normal distribution for all parameters of COP measures. Thus, to evaluate the hypothesized effects of the four tasks (QST, ASC, QST + MT, and ASC + MT) and two planes (ML and AP), repeated analysis of variance (ANOVA) was conducted for all parameters of the COP. Selected pairwise comparisons were explored by using follow-up analyses (Tukey’s test). The level of significance was set at *p* < 0.05.

## 3. Results

### 3.1. COP Variability [mm]

There was a main effect of task (F [3, 69] = 2.86; *p* = 0.043), which indicated the lowest/decreased COP variability in the QST + MT trial. Also, a main effect of plane was observed (F [1, 23] = 15.53; *p* = 0.001) with higher variability in the AP plane. Interestingly, there were no differences between ASC and ASC + MT.

### 3.2. COP Mean Speed [mm/s]

There was a main effect of task (F [3, 69] = 5.16; *p* = 0.003), which reflected decreased COP mean speed in the QST + MT trial. The results of ANOVA proved the main effect of plane (F [1, 23] = 42.87; *p* = 0.000), showing higher values in the AP plane. In addition, a taskc× plane interaction was reported (F [3,69] = 3.49; *p* = 0.020). Post-hoc analysis revealed that performing ASC with an additional mental task resulted in a significantly higher (*p* = 0.003) COP mean speed in the AP plane as compared with QST. There were no differences between ASC and ASC + MT in any plane. In the ML plane, the QST + MT trial showed a significant decrease (*p* = 0.009).

### 3.3. COP Fractal Dimension

There was a main effect of plane (F [1, 23] = 4.360; *p* = 0.048), indicating lower values of the COP fractal dimension in the AP plane. The Tukey’s test pointed at a significant increase of the COP fractal dimension in two trials: ASC (*p* = 0.042) and ASC + MT (*p* = 0.050) compared with QST in the AP plane. However, there were no differences between ASC and ASC + MT trials in any plane. No task main effect (F [3, 69] = 2.481; *p* = 0.068) or task × plane interaction (F [3, 69] = 2.680; *p* = 0.054) showed statistical significance.

### 3.4. COP Sample Entropy

There was a main effect of plane (F [1, 23] = 8.771; *p* = 0.007), showing higher COP sample entropy in the AP plane. No main effect of task (F [3, 69] = 2.236; *p* = 0.092) or task × plane interaction (F [3, 69] = 0.217; *p* = 0.885) was found.

### 3.5. Incremented COP Sample Entropy

There was a main effect of task (F [3, 69] = 4.56; *p* = 0.0057) and plane (F [1, 23] = 176.20; *p* = 0.000). In addition, a task × plane interaction was observed (F [3, 69] = 3.56; *p* = 0.0186). The incremented COP sample entropy increased only during QST + MT (*p* = 0.003) in the ML plane. Post-hoc analysis demonstrated a decrease of the incremented COP sample entropy during ASC + MT (*p* = 0.04) with respect to QST, only in the AP plane. Also, there were no differences between ASC and ASC + MT in the AP or ML planes.

The means (± standard deviation) of all dependent variables are shown in Table 2.

## 4. Discussion

The purpose of this study was to investigate the effects of three additional tasks on postural control in patients with AIS. Four findings seem of particular interest in promoting optimal behaviors in girls with AIS. First, the inclusion of the Stroop test alone resulted in better postural performance, reflected by the decreased COP speed, and in different postural strategies, i.e., increased fractality of the COP and increased entropy of the incremental COP in the ML plane. These changes indicate that an additional mental task had a similar effect on our patients as in healthy young adults [14]. Second, the performance of ASC alone did not affect any spatial measure of the COP nor the entropy of the incremental COP. It did, however, lead to higher values of COP fractality. Both these results seem to account for a good adaptation of our patients to the ASC movements, which not only do not interfere with their postural control but also reflect more adaptive postural strategies. Third, performing the ASC and Stroop test simultaneously preserved the advantageous changes in COP fractality. However, this dual-task resulted in unfavorable changes in COP speed and the incremental COP entropy in the AP plane by increasing and decreasing their values, respectively. It concurs with the results obtained by Huxhold et al. [15], who reported a U-shape relationship between cognitive demand and postural control. While the average or optimal cognitive demand tends to decrease the amplitude of sway, excessive cognitive demand might lead to its increase. And fourth, in spite of the two unfavorable effects caused by performing Stroop test and ASC together in comparison with the baseline data, there were no differences between postural control with ASC compared with ASC and the Stroop test combined.

The different effects of two additional tasks investigated in this study on postural control in AIS are not surprising. The Stroop test is a genuine mental task that is cognitively demanding for unexperienced performers and thus is expected to distract a large part of attention, which is originally devoted to postural control. In contrast, ASC involves strong motor components related to maintaining the recommended body segment positions. While in the initial phase of learning ASC movements, because of their complicated structure, significant attentional resources are necessary; the consecutive training decreases this demand. It is known that practice reduces dual-task interference through increased automaticity [16] or decreased attentional load during learning [16,17].

Consequently, it seems logical that a dual postural task with the Stroop test as an additional mental task should reveal a more automatic mode of postural control [11] due to focusing the attention of AIS on disentangling colors from letters. Indeed, this transition towards increased automaticity was reflected in the present COP speed time-series as higher entropy that was accompanied by a significant decrease in the speed itself in the ML plane. On the other hand, the dual-task, which included the ASC alone, had no effect on the COP amplitude, speed, or entropy, which may be accounted for efficient learning of this task as our subjects had at least three-month experience in practicing ASC. The latter result seems to have a broader significance related to motor control and learning. It shows that it is rather diverted attention from posture and not acquired automaticity [14] that causes changes in the temporal structure of the movement. Such changes concur with the results of other authors [14,18]. In contrast, most previous works did not mention this important difference and tended to equate the contribution of these two processes into the irregularity of sway.

All three tasks, which included ASC and Stroop test separately or as a combined suprapostural task, exhibited a significant increase in the COP fractality in the AP plane. Such an increase in sway complexity is mainly considered as a beneficial change in the control of postural stability. It has been suggested that higher values of the fractal dimension account for the improved overall organization of the postural control system with better use of the available sensory inputs [19]. These changes are associated with better adaptability to new postural requirements, reflecting the ability to use a variety of postural strategies to preserve postural stability [20]. The latter authors documented that balance training, which was focused on a better sensory reweighting, resulted in increased complexity of sway. In the same vein, Casabona et al. [21] showed a larger fractal dimension in ballet dancers than in non-dancers and concluded that this difference might indicate a rearrangement of sensory integration and motor adaptation, necessary with the particular demands of selected ballet performances.

Interestingly, in spite of presumably different attentional demands of the latter three task their individual effects on the COP fractality were very similar. This may indicate limited sensitivity of subjects with AIS to discriminate between the different amounts of attention that was shifted from postural to the suprapostural task. The reason for this limitation may be the foam surface used in the study. Although it is commonly accepted that somatosensory deficit plays a crucial role in postural control in subjects with AIS [22,23], they still seem to strongly rely on proprioception [24]. Simoneau et al. [25] reported that after partial ankle proprioceptive deprivation, returning sensory inputs to normal significantly increased the COP speed. This probably accounted for the difficulty the girls with AIS had with the integration and reweighting of the various afferents to adapt their balance to environmental demands. These problems could be due to the inaccurate identification and transformation of the sensory data into desired postural strategies. A similar situation could have occurred while using a foam pad as a support surface, which by providing distorted or erroneous sensory signals [26] limited the ability of our subjects to shift their attention from posture to the secondary tasks. Though the assessment of the combined effect of ASC and additional mental task on postural control in subjects with AIS was the primary goal of this study, the effects of the mental task alone provide new insights into the potential ability of AIS patients to cope with mental distraction and to follow the instructions received from therapists.

This study should be viewed with some limitations. First, although the Stroop test is often used in the investigation of possible interactions between postural control and secondary mental tasks, it might have been too challenging for subjects with AIS, and its effects on the presented results need to be regarded with caution. The typical activities of daily living are usually familiar to subjects and often include some motor components. While the assessment of the effect of a secondary motor task on postural control is taxing, such an approach would be better suited to answer our research questions. Second, we did not specifically instruct our subjects on which of the two tasks should be prioritized. It is known that the task prioritization may differently affect both tasks, depending on the investigated individuals’ abilities, and the actual results are not easily predictable [27,28]. A similar experiment performed among subjects with AIS may reveal their particular preferences in sharing the attentional resources and help to optimally adjust the therapeutic recommendation to the individual needs.

## 5. Conclusions

The results of this study indicated that postural control of girls with AIS is not susceptible to an adverse interference by an additional mental task. These patients showed good adaptation to the well learned ASC movements, which had no effect on postural behavior. Additionally, an unfamiliar secondary mental task improved their postural control resulting in decreased sway variability and speed with the concomitantly increased irregularity of sway. Finally, when both these tasks were performed collectively, bearing a resemblance to the ADL activities may suggest the need for therapeutic consultation while engaging in more demanding activities.

## Figures and Tables

**Table 1 ijerph-17-01640-t001:** Characteristics of the participants (mean ± s).

Patients with AIS (n = 24)		
Age [years]	13.4 ± 1.6	
Height [cm]	159.5 ± 10.1	
Body mass [kg]	50.8 ± 7.8	
AIS curve pattern	75% R thoracic/L lumbar	25% L thoracolumbar
Primary Cobb angle [degrees]	24.5 ± 7.5	
Risser sign	2.8 ± 0.8	

**Table 2 ijerph-17-01640-t002:** Mean ± standard deviation of mediolateral and anteroposterior center of pressure (COP) outcome measures for the four trials.

Direction	Variable	QST	ASC	QST+MT	ASC+MT
Mediolateral	Variability [mm]	5.35 ± 2.59	4.83 ± 1.71	4.28 ± 1.43	4.94 ± 1.74
	Speed [mm/s]	14.05 ± 3.47	14.55 ± 4.00	12.24 ± 3.18	14.35 ± 4.00
	Fractal dimension	1.43 ± 0.07	1.44 ± 0.05	1.42 ± 0.05	1.42 ± 0.07
	Sample entropy	0.90 ± 0.25	0.96 ± 0.20	0.98 ± 0.19	0.92 ± 0.17
	Incremented sample entropy	1.60 ± 0.16	1.63 ± 0.16	1.72 ± 0.17 *	1.63 ± 0.15
Anteroposterior	Variability [mm]	6.36 ± 2.32	6.07 ± 2.78	5.48 ± 2.07	6.36 ± 2.75
	Speed [mm/s]	16.14 ± 3.93	17.27 ± 5.07	16.27 ± 4.85	18.11 ± 5.75
	Fractal dimension	1.39 ± 0.05	1.42 ± 0.05 *	1.42 ± 0.06 *	1.42 ± 0.05 *
	Sample entropy	0.98 ± 0.25	1.07 ± 0.25	1.11 ±0.23	1.05 ± 0.29
	Incremented sample entropy	1.92 ± 0.13	1.88 ± 0.13	1.93 ± 0.12	1.83 ± 0.15 *

COP—center of pressure; QST: standing upright with a neutral and comfortable stance with the arms relaxed at the sides; ASC: standing upright with autocorrection, where on the “correction” command, the participant performed ASC; QST + MT: QST with a modified color-word Stroop test as an additional mental task; ASC + MT: dual-task: performing ASC with a modified color-word Stroop test as an additional mental task. * Significant difference at *p* < 0.05 compared with QST.

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
