# Peer review of "The Effects of Active Self-Correction on Postural Control in Girls with Adolescent Idiopathic Scoliosis: The Role of an Additional Mental Task"

_ijerph, 2020, doi:10.3390/ijerph17051640_

Round 1

Reviewer 1 Report

Thank you for submitting research work, here are some suggestions from improving the article:

Abstract:

In the method, explain the type of the study. Describe the statistical method used.

Introduction:

Lines 39-41: the sentence is too long; you should add comas.

Line 66: you should avoid statements like: we, in our research, we expected.

Our and we are not used in scientific studies reports. It is better to write :

Present results, this study, it was expected....

Material and Methods:

It is a big fault not to have any control group. Why you don't have it? The results need to be compared to healthy group.

Lines 74-76: It is not clear who did the diagnosis, which method was used.

Lines 74-77: The leg disparency, back pain is not mentioned.

Lines 90-92: Explain why did you choose examination only with eyes open?

The sensorimotor function and proprioception will be better to assess with eyes close?

Line 99: Explain why 5 cm? If some participant has very wide hips?

It is not clear if you performed a prior determination of sample size?

Results:

It is better to design tables so that the P-value can be shown in the table.

Discussion:

Lines 165-183: You replaced in turn 4 research goals and you did not discuss with other results.

The discussion needs to be expanded and further studies are discussed.

Line 210: The latter authors..... this sentence needs stylish correction.

Conclusions:

Lines 247-251: Too long.... it is not clear why did you make a comparison with healthy, physically active young adolescents. You didn't have any control group?  It is a big mistake.

Author Response

Dear Editor,

Thank you very much for your generous comments on the manuscript. All comments were very useful and enabled us to improve the manuscript. Below we present answers and explanations.

Point 1: Abstract: In the method, explain the type of the study. Describe the statistical method used.

Response 1:The abstract was supplemented with the information about statistical method and type of the study (lines 17-18).

Point 2: Introduction: Lines 39-41: the sentence is too long; you should add comas.
Line 66: you should avoid statements like: we, in our research, we expected.
Our and we are not used in scientific studies reports. It is better to write :
Present results, this study, it was expected....

Response 2: We abbreviated the sentence and applied the suggested stylistic corrections to make the text more scientific and understandable to the reader (lines: 41-44, 69)

Point 3: Material and Methods:
It is a big fault not to have any control group. Why you don't have it? The results need to be compared to healthy group.

Response 3: We did not have a control group because performing efficient ASC movement is unfeasible in healthy group.

Point 4: Lines 74-76: It is not clear who did the diagnosis, which method was used.

Response 4: In the Participants (lines: 76-77) information about the person who diagnosed the patients is provided.

Point 5: Lines 74-77: The leg disparency, back pain is not mentioned.

Response 5: We mentioned (lines: 75-77) that diagnosis was adolescent idiopathic scoliosis (AIS) so the leg discrepancy is impossible because the diagnosis would then be functional non-idiopathic scoliosis. We added the information regarding the back pain (line: 79)

Point 6: Lines 90-92: Explain why did you choose examination only with eyes open? The sensorimotor function and proprioception will be better to assess with eyes close?

Response 6: We chose examination with eyes open because the Stroop test requires the participants to see and name the color of the ink of presented words.

Point 7: Line 99: Explain why 5 cm? If some participant has very wide hips?

Response 7: Maybe we used the word standardized incorrectly, and replaced with the word marked (line: 103) The position was fixed for each participants

Point 8: It is not clear if you performed a prior determination of sample size?

Response 8: We wanted to do it but on second thoughts we decided against it. G-power software requires input data concerning the assumed effect size to be detected and the correlations between consecutive measurements. Posturography still suffers from high uncertainty of these data so any subjective assumption may be biased. It seemed better to leave this judgment to the readership. However, similar groups including roughly 20 participants have been investigated in various within-subjects studies satisfying the conditions alfa<0.05 and power >=0.80. We made every effort to exceed this number (our N=24).   

Point 9: Results: It is better to design tables so that the P-value can be shown in the table.

Response 9: We reported the P-values in Table 2 , we used the asterisks * for p<0.05. In Results we showed detailed p-values and we did not want to repeat them again to avoid too much text.

Point 10: Discussion:Lines 165-183: You replaced in turn 4 research goals and you did not discuss with other results.
The discussion needs to be expanded and further studies are discussed.
Line 210: The latter authors..... this sentence needs stylish correction

Response 10: The first paragraph of discussion precisely specifies four findings from our experiment in order to briefly remind the reader of our goals with what was actually achieved. The findings (1) and (2) are collectively expanded in the two next paragraphs, first more generally, and then (in the third paragraph) more precisely.
Due to our omission this last paragraph had no citations in the original submission. In the present revised version we added 4 studies which are helpful in embedding our results in existing knowledge. The two last paragraphs in the Discussion address the remaining findings in relation to findings (1-2).
 We corrected the sentence (lines: 217-218)

Point 11: Conclusions:
Lines 247-251: Too long.... it is not clear why did you make a comparison with healthy, physically active young adolescents. You didn't have any control group?  It is a big mistake.

Response 11: Frankly speaking we were happy to realize that changes in postural control in girls with AIS in the presence of dual tasking were similar to changes in their healthy counterparts reported elsewhere. However, after this remark, we decided to remove this part of the sentence. The conclusion must be limited to our own results.
We did not have a control group because performing efficient ASC movement is unfeasible in healthy group.

Reviewer 2 Report

Although the aim was important. the findings show several limitations that impact on the reliability of the results.

From the most part, the manuscipt has been clearly written, however, the greatest limitation was in the sampling included criteria, since the participant age, onset of AIS and the history of performing ASC these should be considered as three major variables to measure in this study.

Therefore no surprisingly the findings are not particularly novel.

In general, research method process appears to have been appropriately conducted, but the Stroop test which was chosen as the interfere tool also not been explained in the method section why and how? any further evidence-based literature suggested? Thus, the topic is intriguing and has merit, the manuscript is well documented but the findings are not particularly novel.

Author Response

Dear Editor,

Thank you very much for your generous comments on the manuscript. All comments were very useful and enabled us to improve the manuscript. Below we present answers and explanations.

Point 1:

Although the aim was important. the findings show several limitations that impact on the reliability of the results.

From the most part, the manuscipt has been clearly written, however, the greatest limitation was in the sampling included criteria, since the participant age, onset of AIS and the history of performing ASC these should be considered as three major variables to measure in this study.

Response 1: All these variables are very important and impact on results in this study, but this would require a larger group, we strived to have a possible homogeneous group.
Based on the SOSORT (Scientific Society on Scoliosis Orthopaedic and Rehabilitation Treatment) recommendations (Negrini S, Donzelli S, Aulisa AG, Czaprowski D, Schreiber S, de Mauroy JC,  Lebel A. 2018. 2016 SOSORT guidelines: orthopaedic and rehabilitation treatment of idiopathic scoliosis during growth. Scoliosis and Spinal Disorders13(1), 3. ) the article uses the chronological classification of scoliosis according to on the age of the child at which the deformity was diagnosed. According to this classification, the group is homogeneous, all of them are adolescent idiopathic scoliosis. All participants were having received conservative treatment in the form of physiotherapeutic scoliosis-specific exercise (PSSE) for at least three months (lines 77-79) this indicated the same rehabilitation program. We agree that some participants learned ASC faster than others. We had no influence on this.

Point 2: Therefore no surprisingly the findings are not particularly novel.

Response 2: It is difficult to judge how important are this findings. Still our practice as we mentioned in the introduction indicates the dual tasking may interfere with the learning of AIS. So far there is not information regarding this problem in the literature.

Point 3: In general, research method process appears to have been appropriately conducted, but the Stroop test which was chosen as the interfere tool also not been explained in the method section why and how? any further evidence-based literature suggested? Thus, the topic is intriguing and has merit, the manuscript is well documented but the findings are not particularly novel.

Response 3: In the chapter Materials and Methods (lines: 105-111) the Stroop test was explained.

Round 2

Reviewer 1 Report

Addressed paper after corrections has become much more transparent. The authors followed the proposed comments and also answered clearly the questions asked by the reviewer. In my opinion, the paper can be accepted.

Author Response

Dear Reviewer,

Point 1: Addressed paper after corrections has become much more transparent. The authors followed the proposed comments and also answered clearly the questions asked by the reviewer. In my opinion, the paper can be accepted

Response 1: Once again, thank you very much for your hard work and great suggestion, which really improved a value of this manuscript.

Reviewer 2 Report

Thanks to the authors' update information and new evidence to gain this current paper more credibility, in addition, those adding pieces of literature did improve and making the whole paper much clear to read. Few points to make:

  1. line 78-79: 'the participants have received at least PSSE three months', please indicate any upper limit? whether the participants' longer PSSE experience might affect the intervention? 
  2. It is difficult to judge how important are these findings, any small change may determine a great further development, every researcher effort still worth to try to make a better chance for those suffering cases. Therefore this journal encourages and publishes those good evidence-based papers to show all the possibilities.

Author Response

Dear Reviewer,

 Thank you very much for your hard work and great suggestion, which really improved value of this manuscript.

We believe that the manuscript is now suitable for publication.

Point 1: line 78-79: 'the participants have received at least PSSE three months', please indicate any upper limit? whether the participants' longer PSSE experience might affect the intervention?

Response 1: Thank you for your comment, this information is very valuable and was added in line 78. We consider that different times of PSSE experience might affect the intervention. This is interesting for a future research and may be helpful in clinical practice.

Point 2: It is difficult to judge how important are these findings, any small change may determine a great further development, every researcher effort still worth to try to make a better chance for those suffering cases. Therefore this journal encourages and publishes those good evidence-based papers to show all the possibilities

Response 2: We absolutely agree with you. Once again, thank you very much for your generous comments on the manuscript. All comments were very useful and enabled us to improve the manuscript.

M. Sc. Elżbieta Piątek
Faculty of Physiotherapy, University School of Physical Education
